# Effects of vitamin D on COVID-19 risk and hospitalisation in the UK biobank

**Maria J. Monroy-Iglesias**[1]*, **Rathesh Thavarajah**[1], **Kerri Beckmann**[2], **Debra H. Josephs**[1,3], **Sheeba Irshad**[3,4,5], **Sophia N. Karagiannis**[5,6], **Mieke Van Hemelrijck**[1], **Aida Santaolalla**[1]

**1** Transforming Outcomes through Research (TOUR), Centre for Cancer, Society, and Public Health, School of Cancer and Pharmaceutical Sciences, King's College London, London, United Kingdom, **2** Cancer Epidemiology and Population Health Research Group, University of South Australia, Adelaide, Australia, **3** Department of Medical Oncology, Guy's and St Thomas' NHS Foundation Trust, London, United Kingdom, **4** School of Cancer and Pharmaceutical Sciences, King's College London, London, United Kingdom, **5** Breast Cancer Now Research Unit, School of Cancer & Pharmaceutical Sciences, King's College London, London, United Kingdom, **6** St. John's Institute of Dermatology School of Basic & Medical Biosciences, & KHP Centre for Translational Medicine, King's College London, London, United Kingdom

* maria.j.monroy_iglesias@kcl.ac.uk

## Abstract

### Background

Vitamin D (VitD) plays an important role in immune modulation. VitD deficiency is associated with increased susceptibility to acute respiratory syndrome as observed in COVID-19. We evaluated potential associations between serum VitD levels and risk of COVID-19 infection and hospitalisation, within the overall and cancer populations.

### Methods

We performed a nested case-control study within the UK biobank cohort, among all individuals with at least one serum VitD level measurement at baseline (2006−2010) and a COVID-19 polymerase chain reaction (PCR) results recorded, and individuals with previous cancer diagnosis. Binary multivariable logistic regression was performed to assess associations between VitD levels and risk of COVID-19 infection (positive PCR), and hospitalisation (COVID-19-positive PCR in hospital), and stratified by ethnicity.

### Results

Of 151,543 participants, 21,396 tested positive for COVID-19. Of 24,400 individuals with cancer, 2,608 tested positive. In the total cohort, VitD insufficiency (Adjusted Odds Ratio (aOR) 0.97, 95% Confidence Interval (CI) 0.94–1.00) and deficiency (aOR 0.95, 95%CI 0.90–0.99) were associated with slightly lower odds of COVID-19 infection. In contrast, both VitD insufficiency (aOR 1.19, 95%CI 1.08–1.31) and deficiency (aOR 1.36, 95%CI 1.19–1.56) were associated with higher odds of COVID-19

provided the original author and source are credited.

**Data availability statement:** Data cannot be shared publicly due to the ethical and legal restrictions of the UK Biobank. Data are available through application to the UK Biobank (www.ukbiobank.ac.uk) for researchers who meet the criteria for access to confidential data.

**Funding:** The author(s) received no specific funding for this work.

**Competing interests:** The authors have declared that no competing interests exist.

hospitalisation. Among Asian (aOR 1.50; 95%CI 1.08–2.07) and Black (aOR 1.57; 95%CI 1.14–2.16) participants, VitD deficiency was associated with higher odds of COVID-19 infection. Among White participants, VitD insufficiency was associated with slightly lower odds of COVID-19 infection (aOR 0.97; 95%CI 0.86–0.95), while both VitD insufficiency (aOR 1.19; 95%CI 1.08–1.32) and deficiency (aOR 1.44; 95%CI 1.25–1.66) were associated with increased odds of hospitalisation. In the cancer population, vitamin D deficiency was associated with higher odds of infection only among Black participants (aOR 3.50; 95%CI 1.22–10.01); no other associations were observed.

## Conclusions

Low VitD levels were associated with an increased risk of COVID-19 hospitalisation but showed only a weak association with infection risk. Black and Asian populations had higher infection risk associated with VitD deficiency, but this did not translate to increased hospitalisation. In contrast, White populations with low VitD levels exhibited a higher risk of hospitalisation. There was no evidence of an interaction between VitD levels and ethnicity affecting infection or hospitalisation risk. In the cancer cohort, no significant associations were observed for COVID-19 infection or hospitalisation.

## Introduction

The global impact of COVID-19 on healthcare services has been significant, fuelling extensive research interest in exploring potential pathophysiological mechanisms underlying the disease [1]. At the start of the pandemic, several studies were conducted looking at risk factors influencing COVID-19 infection rates, severity, and mortality. These studies consistently identified age, male sex, smoking status, obesity, belonging to certain ethnic groups, and a compromised immune system, among others, as factors contributing to an increased likelihood of developing severe disease and increased mortality rates [2,3].

Vitamin D (VitD) plays a crucial role in modulating both the innate and adaptive immune response [4]. Deficiency in VitD has been associated with increased susceptibility to respiratory tract infections [5–7] and is recognised as a risk factor for the development of exaggerated and persistent inflammation, which acts as a precursor to acute respiratory distress syndrome (ARDS) [8]. Consequently, since the outset of the pandemic, several studies have investigated the relationship between VitD and risk of COVID-19 [9]. There is growing evidence of the association between VitD deficiency and risk of COVID-19 infection and severity [10–12]. A recent review reported consistent findings of a protective effect of higher VitD levels against both infection and severe disease [13]. Additionally, a systematic review and meta-analysis of 23 studies found that VitD supplementation had no

significant impact on infection risk but showed a protective effect against severe disease and mortality [14]. Low VitD levels have been associated with older age, smoking, obesity, chronic illnesses (e.g., diabetes, cancer, hypertension), and belonging to certain ethnic groups [15], many of which are also established risk factors of COVID-19 infection and severity [2,3]. In particular, individuals from Black, Asian, and minority ethnic backgrounds often have lower levels of VitD due to reduced synthesis from ultraviolet radiation (UVR) [16], a disparity attributed to differences in skin melanin content [17].

Although associations between VitD status and COVID-19 outcomes have been examined in the general population, there remains a notable gap in research focusing on individuals with cancer. Given VitD's potential protective role in cancer—through mechanisms such as immune modulation and regulation of cell growth [18–21]—it is important to consider how VitD status may influence COVID-19 outcomes in this group. Understanding the potential link between VitD and COVID-19 within the cancer population could inform clinical practice and public health strategies aimed at mitigating COVID-19 morbidity and mortality in this vulnerable group.

The current study aimed to investigate the association between VitD levels and both COVID-19 infection and hospitalisation using data from the prospective UK Biobank. Additionally, we examined these associations across the general population, a subgroup of cancer patients, and considered potential variations by ethnicity.

## Materials and methods

### Study population

Our study population consisted of participants enrolled in the UK Biobank, a prospective cohort study comprising of approximately 500,000 participants aged 40–69 years, registered with the UK National Health Service (NHS), at the time of their initial assessment. All participants were recruited between 2006–2010. Participants attended one of the assessment centres across England, Wales, and Scotland, where they were interviewed, underwent physical examinations, and had biological samples collected. Their data have been linked with consent, to various other data sources including cancer registry, national death registrations, hospital separation records and COVID testing results. A more detailed description of the UK Biobank cohort can be found elsewhere [22,23].

In the current study, two cohorts from UK Biobank were included. For our first cohort, eligible participants were men and women aged 40–69 years at assessment, who had a serum VitD baseline measurement, were alive during the COVID-19 pandemic, and had at least one COVID-19 polymerase chain reaction (PCR) result. Participants who did not meet the above criteria were excluded from the study, resulting in a final study sample of 151,543 individuals. For our second cohort, we included a subset of eligible participants who were also aged 40−69 years at assessment, had a serum VitD baseline measurement, were alive during the COVID-19 pandemic, had at least one PCR test, and had a cancer diagnosis prior to 2019. Participants with a cancer diagnosis after December 31st 2019 were excluded from the study cohort, resulting in a final study sample of 24,400 individuals.

UK Biobank data were collected in accordance with the UK Biobank Ethics and Governance Council (https://www.ukbiobank.ac.uk/learn-more-about-uk-biobank/about-us/ethics). All participants gave broad consent to use of their anonymized data and samples for any health-related research, and this study was approved by the North West Multicentre Research Ethics Committee in the United Kingdom (Ref: 11/NW/0274). All participants provided written informed consent.

### Vitamin D assessment

The main exposure variable in our analysis was endogenous VitD status. Blood samples were collected from which biochemical assays were conducted to measure serum 25(OH)D levels (nmol/L). Serum 25(OH)D (VitD) (nmol/L) was measured at the baseline assessment which was conducted between 2006–2010. For samples below the assay's detectable

range, a value of 10 nmol/L was assigned, while values exceeding the assay's upper threshold were capped at 375 nmol/L to reflect plausible physiological levels.

A small subset of the cohort had a second VitD measurement taken during the first repeat assessment visit between 2012 and 2013 (n = 17,036). For individuals with two measurements, the mean VitD level was calculated. As the two measurements were highly correlated, averaging them provided a stable estimate of typical VitD status. For the purposes of the analyses, VitD was categorized into three groups based on the Public Health England definitions: Deficiency (0–< 24 nmol/L), Insufficiency (25–50 nmol/L), and Normal (>50 nmol/L).

### COVID-19 assessment

Two endpoints were investigated in the current study, first COVID-19 Infection which is measured using PCR COVID-19 test result (i.e., positive, negative). In individuals who had multiple positive test results, the first positive test result was included in the analysis. The second COVID-19 outcome we analysed was hospitalisation which was measured using Hospital Admission (inpatient or outpatient). Using the origin, the facility type, or the patient type of the COVID-19 test result data we were able to identify whether the individual was an inpatient when the test was taken. PCR test results were available from March through October 2020. Data measured at baseline from the UK Biobank was linked to COVID-19 test result and hospitalisation data from England, Scotland and Wales provided by Public Health England (PHE), Public Health Scotland (PHS) and Secured Anonymised Information Linkage (SAIL).

### Data analysis

Descriptive statistical analysis of both cohorts was conducted. Binary logistic regression was used to calculate odds ratios (OR) and 95% confidence intervals (CI) for risk of COVID-19 infection and hospitalisation. The following covariates (assessed at baseline) were included as confounders: Sex (Male, Female), age at entry in the UK Biobank cohort (<40, 40–48, 50–59, 60–69), socioeconomic status (SES, Townsend Deprivation Index [24], from 1 (most affluent) to 5 (most deprived)), overall health rating (excellent, good, fair, poor), body mass index (BMI, underweight, normal, overweight, obese), smoking status (never, former, current), and alcohol use (never, occasional, 1–3 per month, 1–2 per week, 3–4 per week, daily). Data on Ethnicity were collected at baseline via self-reporting by participants, using touch-screen questionnaires. In our current study, the 20 categories recorded in the UK Biobank were simplified into five groups (i.e., White, Mixed, Asian, Black, Other Ethnic Background; S1 Table). Guided by a directed acyclic graph (DAG) to assess the relationships between potential confounders, we developed two models to examine the relationship between vitD levels and risk of COVID-19 infection and hospitalisation. Our adjusted Model A included: sex, age, SES, overall health rating, BMI, and smoking status. Model B included all factors in Model A, in addition to Ethnicity.

Furthermore, given that VitD levels are known to vary by Ethnicity [25], we conducted a series of sensitivity analyses to assess the possibility of effect modification and exclude associations potentially due to Ethnicity. To address this, we performed stratification analyses by ethnicity using Model A to assess the association between VitD and COVID-19 infection and hospitalisation, in both cohorts.

Analyses were performed using IBM SPSS Statistics version 27.0.1.0 and using Stata IC 15.1 (College Station, TX, USA). All statistical tests were two-sided, with p-values <0.05 considered to be statistically significant.

### Results

A total of 21,396 (14%) participants from the general cohort had at least one positive COVID-19 test result, of whom 2,376 (11%) were hospitalized with to COVID-19. In the cancer cohort, a total of 2,608 (11%) participants had at least one positive COVID-19 test result and 374 (14%) were hospitalized with COVID-19. Participant characteristics for both cohorts are shown in Tables 1 and 2.

**Table 1. Characteristics of the study population by positive or negative COVID-19 test result.**

| | Positive test for COVID-19 n = 21,396 (14%) | Negative test for COVID-19 n = 130,147 (86%) | Total Population n = 151,543 | P Value |
|---|---|---|---|---|
| **Previous Cancer Diagnosis** | | | | <0.001 |
| Yes | 2,608 (12%) | 21,792 (17%) | 24,400 (16%) | |
| No | 18,788 (88%) | 108,355 (83%) | 127,143 (84%) | |
| **Sex** | | | | 0.007 |
| Male | 10,147 (47%) | 60,440 (46%) | 70,587 (47%) | |
| Female | 11,249 (53%) | 69,707 (54%) | 80,956 (53%) | |
| **Age (Years)** | | | | <0.001 |
| Mean (SD) | 53 (8) | 57 (8) | 56 (8) | <0.001 |
| Under 40 | 492 (2%) | 1,357 (1%) | 1,849 (1%) | |
| 40-49 | 8,944 (42%) | 32,104 (25%) | 41,048 (27%) | |
| 50-59 | 6,811 (32%) | 45,220 (35%) | 52,031 (34%) | |
| 60-69 | 5,149 (24%) | 51,466 (39%) | 56,615 (37%) | |
| **Body Mass Index (BMI) (kg/m²)** | | | | <0.001 |
| Mean | 28 (5) | 28 (5) | 28 (5) | <0.001 |
| Underweight (<18.5) | 79 (0.4%) | 588 (0.5%) | 667 (0.4%) | |
| Healthy Weight (18.5–24.9) | 6,099 (28%) | 40,993 (31%) | 47,092 (31%) | |
| Overweight (25.0–29.9) | 9,134 (43%) | 55,155 (42%) | 64,289 (42%) | |
| Obese ( ≥ 30) | 6,008 (28%) | 32,913 (25%) | 38,921 (26%) | |
| Missing | 76 (0.4%) | 498 (0.4%) | 574 (0.4%) | |
| **Ethnic Background** | | | | <0.001 |
| White | 19,615 (92%) | 123,583 (95%) | 143,198 (94%) | |
| Mixed | 162 (1%) | 755 (0.6%) | 917 (1%) | |
| Asian or Asian British | 765 (4%) | 2,316 (2%) | 3,081 (2%) | |
| Black or Black British | 511 (2%) | 1853 (1%) | 2364 (2%) | |
| Other ethnic group | 252 (1%) | 1,102 (1%) | 1354 (1%) | |
| Missing | 91 (0.4%) | 538 (0.4%) | 629 (0.4%) | |
| **Townsend Deprivation Index (Quintiles)** | | | | <0.001 |
| 1 (most deprived) | 3,631 (17%) | 26,638 (21%) | 30,629 (20%) | |
| 2 | 3,877 (18%) | 26,394 (20%) | 30,721 (20%) | |
| 3 | 4,314 (20%) | 25,954 (20%) | 30,628 (20%) | |
| 4 | 4,521 (21%) | 25,747 (20%) | 30,628 (20%) | |
| 5 (least deprived) | 5,022 (24%) | 25,247 (19%) | 30,629 (20%) | |
| **Current Smoking Status** | | | | 0.02 |
| Yes, on all or most days | 1,510 (7%) | 9,638 (7%) | 11,148 (7%) | |
| Yes, occasionally | 697 (3%) | 3,818 (3%) | 4,515 (3%) | |
| No | 19,149 (89%) | 116,467 (89%) | 135,616 (89%) | |
| Missing | 40 (0.2%) | 224 (0.2%) | 164 (0.2%) | |
| **Frequency of alcohol intake** | | | | <0.001 |
| Daily or almost daily | 3,540 (16%) | 28,202 (22%) | 31,742 (21%) | |
| Three/Four times a week | 4,871 (23%) | 30,760 (24%) | 35,631 (23%) | |
| Once/Twice a week | 6,178 (29%) | 33,260 (26%) | 39,438 (26%) | |
| One to three times a month | 2,608 (12%) | 13,857 (11%) | 16,465 (11%) | |
| Special occasions only | 2,425 (11%) | 14,172 (11%) | 16,597 (11%) | |

*(Continued)*

Table 1. (Continued)

| | Positive test for COVID-19 n = 21,396 (14%) | Negative test for COVID-19 n = 130,147 (86%) | Total Population n = 151,543 | P Value |
|---|---|---|---|---|
| Never | 1,732 (8%) | 9,614 (7%) | 11,346 (7%) | |
| Missing | 42 (0.2%) | 282 (0.2%) | 324 (0.2%) | |
| **Overall Health Rating** | | | | <0.001 |
| Poor | 958 (4%) | 6,347 (5%) | 7,305 (5%) | |
| Fair | 4,794 (22%) | 28,239 (22%) | 33,033 (22%) | |
| Good | 12,302 (57%) | 73,873 (57%) | 86,175 (57%) | |
| Excellent | 3,203 (15%) | 20,982 (16%) | 24,185 (16%) | |
| Missing | 139 (0.5%) | 706 (0.5%) | 845 (0.5%) | |
| **Vitamin D (nmol/L)** | | | | <0.001 |
| Mean (SD) | 48.50 (21) | 49.70 (21) | 49.53 (21) | <0.001 |
| Deficient (<25nmol/L) | 3,080 (14%) | 16,278 (12%) | 19,358 (13%) | |
| Insufficient (25–50nmol/L) | 8,930 (42%) | 53,509 (41%) | 62,439 (41%) | |
| Normal (>50nmol/L) | 9,386 (44%) | 60,360 (46%) | 69,746 (46%) | |

## Overall population

The results from our crude analysis showed that both VitD insufficiency (OR 1.07; 95%CI 1.04–1.10) and deficiency (OR 1.21; 95%CI 1.16–1.27) were associated with an increased risk of COVID-19 infection, compared with normal VitD levels. This association was not significant when adjusting for confounding factors in Model A. However, when further adjusting for Ethnicity (Model B), a borderline decreased risk of COVID-19 infection was observed for both insufficiency (OR 0.97; 95%CI 0.94–1.00) and deficiency (OR 0.95; 95%CI 0.90–0.99) (Table 3). In contrast, a consistent association was found for insufficient and deficient VitD levels and risk of COVID-19 hospitalisation. Results from Model A showed an increased risk of COVID-10 hospitalisation for both insufficiency (OR 1.19, 95%CI 1.08–1.31) and deficiency (OR 1.38, 95%CI 1.21–1.57). This association remained statistically significant after further adjusting for Ethnicity (Model B), for both insufficiency (OR 1.19; 95%CI 1.08–1.31) and deficiency (OR 1.36; 95%CI 1.19–1.56) (Table 4).

Results of our stratified analyses in the overall population are shown in S2 and S3 Tables. The previously observed borderline negative association between both VitD insufficiency and deficiency with COVID-19 infection remained within the White population (Insufficient OR 0.97, 95%CI 0.86–0.95; Deficient OR 0.90, 95%CI 0.86–0.95). However, VitD deficiency was significantly associated with an increased risk of COVID-19 infection in the Asian (OR 1.50; 95%CI 1.08–2.07) and Black (1.57; 95%CI 1.14–2.16) population. No significant associations were found for VitD insufficiency and COVID-19 risk in the Asian or Black population. When considering COVID-19 hospitalisation, the previously observed increased risk in VitD insufficiency and deficiency remained in the White population (Insufficient OR 1.19; 95%CI 1.08–1.32; Deficient OR 1.44; 95%CI 1.25–1.66). No significant associations were found between COVID-19 hospitalisation in either the Asian or Black population.

## Cancer population

Results from our crude analysis showed a borderline significant association for both VitD insufficiency (OR 1.10, 95%CI 1.01–1.20) and deficiency (OR 1.17, 95%CI 1.02–1.34) and risk of COVID-19 infection. However, both associations were lost for Models A and B (Table 5). Similarly, results from our crude analysis showed that both VitD insufficiency (OR 1.28;

**Table 2. Characteristics of the cancer population by positive or negative COVID-19 test result.**

| | Positive test for COVID-19 n = 2,608 (10.7%) | Negative test for COVID-19 n = 21,792 (89.3%) | Total Population n = 24,400 | P Value |
|---|---|---|---|---|
| **Sex** | | | | 0.003 |
| Male | 1,119 (43%) | 10,014 (46%) | 11,133 (46%) | |
| Female | 1,489 (57%) | 11,778 (54%) | 13,267 (54%) | |
| **Age (Years)** | | | | <0.001 |
| Mean (SD) | 57 (8) | 60 (7) | 60 (7) | <0.001 |
| Under 40 | 26 (1%) | 94 (0.4%) | 120 (0.5%) | |
| 40-49 | 613 (24%) | 2,707 (12%) | 3,320 (14%) | |
| 50-59 | 897 (34%) | 6,650 (30%) | 7,547 (30%) | |
| 60-69 | 1,072 (41%) | 12,341 (57%) | 13,413 (55%) | |
| **Body Mass Index (BMI) (kg/m²)** | | | | 0.02 |
| Mean (SD) | 28 (5) | 27 (5) | 27 (5) | <0.001 |
| Missing | 8 (0.3%) | 80 (0.4%) | 88 (0.4%) | |
| Underweight (<18.5) | 11 (0.4%) | 112 (0.5%) | 123 (0.5%) | |
| Healthy Weight (18.5–24.9) | 752 (29%) | 6,851 (31%) | 7,603 (31%) | |
| Overweight (25.0–29.9) | 1,126 (43%) | 9,357 (43%) | 10,483 (43%) | |
| Obese ( ≥ 30) | 711 (27%) | 5,392 (25%) | 6103 (25%) | |
| **Ethnic Background** | | | | <0.001 |
| Missing | 11 (0.4%) | 78 (0.4%) | 89 (0.4%) | |
| White | 2,492 (96%) | 21,181 (97%) | 23,676 (97%) | |
| Mixed | 16 (1%) | 86 (0.4%) | 102 (0.4%) | |
| Asian or Asian British | 29 (1%) | 171 (1%) | 200 (1%) | |
| Black or Black British | 42 (2%) | 188 (1%) | 230 (1%) | |
| Other ethnic group | 18 (1%) | 88 (0.4%) | 106 (0.4%) | |
| **Townsend Deprivation Index (Quintiles)** | | | | <0.001 |
| Mean (SD) | −1 (3) | −2 (3) | −1 (3) | <0.001 |
| 1 (most deprived) | 507 (19%) | 4,706 (22%) | 5,213 (21%) | |
| 2 | 498 (19%) | 4,740 (22%) | 5,238 (21%) | |
| 3 | 526 (20%) | 4,434 (20%) | 4,960 (20%) | |
| 4 | 521 (20%) | 4,164 (19%) | 4,685 (20%) | |
| 5 (least deprived) | 553 (21%) | 3,723 (17%) | 4,276 (18%) | |
| **Current Smoking Status** | | | | 0.83 |
| No | 2,352 (90%) | 19,748 (91%) | 22,100 (91%) | |
| Yes, on all or most days | 180 (7%) | 1,476 (7%) | 1,656 (7%) | |
| Only occasionally | 72 (3%) | 543 (2%) | 615 (3%) | |
| **Frequency of alcohol intake** | | | | <0.001 |
| Daily or almost daily | 476 (18%) | 5,135 (24%) | 5,611 (23%) | |
| Three/Four times a week | 630 (24%) | 5,105 (23%) | 5,735 (22%) | |
| Once/Twice a week | 775 (30%) | 5,441 (25%) | 6216 (25%) | |
| One to three times a month | 287 (11%) | 2,205 (10%) | 2,492 (10%) | |
| Special occasions only | 256 (10%) | 2,324 (11%) | 2,580 (11%) | |
| Never | 179 (7%) | 1,514 (7%) | 1,720 (7%) | |
| **Overall Health Rating** | | | | 0.72 |
| Poor | 146 (6%) | 1,171 (5%) | 1,317 (5%) | |
| Fair | 639 (24%) | 5,173 (24%) | 5,812 (24%) | |

*(Continued)*

| | Positive test for COVID-19 n = 2,608 (10.7%) | Negative test for COVID-19 n = 21,792 (89.3%) | Total Population n = 24,400 | P Value |
|---|---|---|---|---|
| Good | 1,451 (56%) | 12,224 (56%) | 13,675 (56%) | |
| Excellent | 353 (14%) | 3,099 (14%) | 3,452 (14%) | |
| **Vitamin D (nmol/L)** | | | | 0.01 |
| Mean (SD) | 51 (21) | 52 (21) | 52 (21) | 0.59 |
| Deficient (<25nmol/L) | 306 (12%) | 2,307 (10%) | 2,613 (10%) | |
| Insufficient (25–50nmol/L) | 1,047 (40%) | 8,386 (39%) | 9,433 (39%) | |
| Normal (>50nmol/L) | 1,255 (48%) | 11,099 (51%) | 12,354 (51%) | |

**Table 3. Odds Ratios for the association between Vitamin D status and COVID-19 infection, within the overall population.**

| Vitamin D status | Unadjusted | | | Adjusted – Model A [a] | | | Adjusted – Model B [b] | | |
|---|---|---|---|---|---|---|---|---|---|
| | OR | 95% CI | p-value | OR | 95% CI | p-value | OR | 95%CI | p-value |
| **Normal** | reference | – | – | reference | – | – | reference | – | – |
| **Insufficient** | 1.07 | 1.04-1.10 | <0.001 | 0.99 | 0.97-1.06 | 0.49 | 0.97 | 0.94-1.00 | 0.07 |
| **Deficient** | 1.21 | 1.16-1.27 | <0.001 | 1.01 | 0.95-1.01 | 0.36 | 0.95 | 0.90-0.99 | 0.04 |

[a]Model A- Adjusted for Sex, Age at recruitment, Townsend Deprivation Index, overall health rating, BMI, and smoking status.

[b]Model A + ethnicity.

**Table 4. Odds Ratios for the association between Vitamin D status and COVID-19 hospitalisation, within the overall population.**

| Vitamin D status | Unadjusted | | | Adjusted – Model A [a] | | | Adjusted – Model B [b] | | |
|---|---|---|---|---|---|---|---|---|---|
| | OR | 95% CI | p-value | OR | 95% CI | p-value | OR | 95%CI | p-value |
| **Normal** | reference | – | – | reference | – | – | reference | – | – |
| **Insufficient** | 1.22 | 1.11-1.34 | <0.001 | 1.19 | 1.08-1.31 | <0.001 | 1.19 | 1.08-1.31 | <0.001 |
| **Deficient** | 1.42 | 1.26-1.61 | <0.001 | 1.38 | 1.21-1.57 | <0.001 | 1.36 | 1.19-1.56 | <0.001 |

[a]Model A- Adjusted for Sex, Age at recruitment, Townsend Deprivation Index, overall health rating, BMI, and smoking status.

[b]Model A + ethnicity.

**Table 5. Odds Ratios for the association between Vitamin D status and COVID-19 infection, within the cancer population.**

| Vitamin D status | Unadjusted | | | Adjusted – Model A [a] | | | Adjusted – Model B [b] | | |
|---|---|---|---|---|---|---|---|---|---|
| | OR | 95% CI | p-value | OR | 95% CI | p-value | OR | 95%CI | p-value |
| **Normal** | reference | – | – | reference | – | – | reference | – | – |
| **Insufficient** | 1.10 | 1.01-1.20 | 0.02 | 1.03 | 0.94-1.12 | 0.48 | 1.02 | 0.93-1.12 | 0.57 |
| **Deficient** | 1.17 | 1.02-1.34 | 0.01 | 1.00 | 0.88-1.15 | 0.89 | 0.98 | 0.85-1.13 | 0.84 |

[a]Model A- Adjusted for Sex, Age at recruitment, Townsend Deprivation Index, overall health rating, BMI, and smoking status.

[b]Model A + ethnicity.

95%CI 1.02–1.62) and deficiency (OR 1.45, 95%CI 1.03–2.04) increased the risk of COVID-19 hospitalisation (Table 6). However, these associations did not persist in the adjusted analyses.

Results from our stratification analysis are shown in S4 and S5 Tables. We found no significant associations between VitD and for COVID-19 infection or hospitalisation, within any of the ethnicity subgrouping.

**Table 6. Odds Ratios for the association between Vitamin D status and COVID-19 hospitalisation, within the cancer population.**

| Vitamin D status | Unadjusted | | | Adjusted – Model A [a] | | | Adjusted – Model B [b] | | |
|---|---|---|---|---|---|---|---|---|---|
| | OR | 95% CI | p-value | OR | 95% CI | p-value | OR | 95%CI | p-value |
| Normal | reference | – | – | reference | – | – | reference | – | – |
| Insufficient | 1.28 | 1.02-1.62 | 0.04 | 1.22 | 0.96-1.56 | 0.10 | 1.22 | 0.95-1.55 | 0.11 |
| Deficient | 1.45 | 1.03-2.04 | 0.03 | 1.30 | 0.91-1.86 | 0.14 | 1.29 | 0.89-1.87 | 0.16 |

[a]Model A- Adjusted for Sex, Age at recruitment, Townsend Deprivation Index, overall health rating, BMI, and smoking status.
[b]Model A+ethnicity.

## Discussion

We presented a comprehensive study investigating the association between VitD and COVID-19 infection and hospitalisation in the UK Biobank cohort and a subset of participants with a cancer diagnosis prior to 2019. In our prospective study of 21,396 participants with COVID-19 test results, we found no evidence of increased risk of COVID-19 infection among individuals with VitD insufficiency or deficiency in the general (predominantly White) population. However, our results suggest that VitD insufficiency and deficiency are associated with an increased risk of COVID-19 hospitalisation. No significant associations were observed between VitD levels and risk of COVID-19 infection or hospitalisation in the subset of participants with a prior diagnosis of cancer.

Results of prior studies have been inconsistent [8,26]. A meta-analysis by Pereira et al, which included 26 prospective studies that assessed VitD deficiency and risk of COVID-19 infection, severity, hospitalisation, and death, reported a positive association between VitD deficiency and severe COVID-19 (OR 1.65: 95%CI 1.30–2.09), hospitalisation (OR 1.81; 95%CI 1.42–2.21), and death (OR 1.82; 1.06–2.58). However, they found no significant association between VitD deficiency and COVID-19 infection [27]. Definitions of deficiency in the included studies ranged from <30 nmol/L to <50 nmol/L. Our current study findings of no significant association between VitD levels and risk of COVID-19 infection, but a positive relationship between VitD deficiency and insufficiency, and risk of COVID-19 hospitalisation, are in line with the findings from that meta-analysis. However, they are not consistent with the findings from another meta-analysis by Teshome et al which reported a significantly increased risk of COVID-19 infection (OR 1.80; 95%CI 1.72–1.88) with VitD deficiency (defined as <50nmol/L) [12].

Various other studies have also used data from the UK Biobank to examine the association between VitD and risk of COVID-19 infection and severe disease [28–32]. Hastie et al, and Raisi-Estabragh et al reported no significant associations between VitD levels and COVID-19 infection and concluded that differences in VitD levels did not explain ethnic differences in COVID-19 infection rates [29,30]. Hastie et al, and Lin et al found no associations between VitD levels and risk of COVID-19 hospitalisation and mortality [31,32]. The difference between our findings and the findings of these studies with regard to COVID-19 hospitalisation and severity may be attributed to the variations in statistical methods, with both of the other studies adjusting for additional clinical variables. Additionally, Lin et al. reported their findings based on VitD levels during British summertime and adjusted for ethnicity, UK region which may have led to the variation from our analysis [32]. Lastly, a study by Ma et al. looking at VitD supplementation suggested that habitual use of supplements was associated with a lower risk of COVID-19 infection [33]. All studies used the same definitions for VitD deficiency (<25 nmol/L) and insufficiency (<50 nmol/L) as our study.

Although epidemiological evidence on the link between VitD and COVID-19 severity remains inconsistent, there are strong biological reasons to support a potential connection. VitD plays a crucial regulatory role in both acquired and innate immunity, and higher levels have been associated with a reduced risk of the cytokine storm and subsequent acute respiratory distress syndrome (ARDS), a leading cause of COVID-19 mortality [34,35]. Two key mechanisms by which VitD may

exert protective effects include the upregulation of antimicrobial peptides such as cathelicidin (LL-37), which reduce viral and bacterial viability, and the suppression of pro-inflammatory cytokines that contribute to disease progression. These effects are likely most beneficial early in the course of infection [36]. Importantly, clinical studies have highlighted the complexity of COVID-19 progression, including evidence that shows delays between symptom onset and hospital admission which may in turn influence patient outcomes [37,38]. Furthermore, acute inflammation itself may reduce circulating 25(OH)D concentrations, complicating the interpretation of VitD measurements taken during or shortly after illness onset [39,40]. In our study, no significant association was observed between VitD levels and COVID-19 infection or hospitalization among participants with a prior cancer diagnosis. This lack of association may be due to immune suppression from cancer treatments, such as chemotherapy, which could blunt the immune-modulating effects of VitD [41]. Alternatively, the chronic inflammation and immune dysfunction commonly seen in cancer patients might overshadow any potential protective benefits of adequate VitD levels [42]. Further studies are needed to disentangle the interplay between cancer, immune function, and VitD in the context of COVID-19.

Results from our sensitivity analyses suggest that Black and Asian ethnicities with deficient VitD levels are at increased risk of testing positive for COVID-19. However, no significant associations were found between VitD deficient or insufficient ethnic minorities (i.e., Asian, Black, Mixed, or Other) and COVID-19 hospitalisation. These observations are only partially consistent with other studies which have reported greater susceptibility to both infection and severe presentations of COVID-19 among ethnic groups [43,44]. A meta-analysis including 68 studies reported that COVID-19 positivity and intensive care unit (ICU) admission rates were higher in African American, Hispanic, and Asian American individuals compared with White individuals [45]. Explanations for increased susceptibility to COVID-19 infection among ethnic minority groups may include increased socio-economic deprivation, cultural factors, greater levels of comorbidity and lower levels of Vit D due to darker skin pigmentation [45,46]. In a meta-analysis by Megesh et al, the increased risk among different racial groups was attenuated when adjusting for sociodemographic characteristics (e.g., deprivation index), but remained higher compared the White population [45]. Higher prevalence of comorbidities in some minority populations, such as a higher rate of hypertension in the Black population [47] may be an important contributing factor underlying the association with ethnicity. Moreover, VitD levels in people from minority ethnic groups tend to be decreased due to darker skin pigmentation [16,17]. Higher levels of melanin pigment, which protect underlying skin from damage caused by sun exposure, lead to reduced UVR available for VitD synthesis of the skin and therefore, lower VitD levels [16]. Furthermore, results from a study published at the beginning of the pandemic which also analysed data from the UK Biobank reported that VitD was not associated with risk of COVID-19 infection. The same study also found that Black and Asian ethnicities were associated with an increased risk of COVID-19 infection, and adjustment for VitD levels made little difference to the magnitude of the associations [30]. These findings suggested that despite VitD levels being lower in minority ethnic populations, other socioeconomic, cultural or health-related factors are more likely to explain the association between ethnicity and COVID-19 infection.

To our knowledge, this is the first study to analyse the association between VitD and risk of COVID-19 within the cancer population. Several studies have demonstrated that cancer patients are at a much higher risk of adverse COVID-19 outcomes (e.g., severe COVID-19, hospital admission, and morality) [18,48,49]. However, in our cancer cohort (some of whom may be immunocompromised), no significant associations between VitD levels and risk of COVID-19 infection or hospitalisation were found. These findings suggest that any impact VitD might have on risk of COVID-19 infection or hospitalisation is no greater among cancer survivors.

Lastly, various studies have analysed whether VitD supplementation has an effect on COVID-19 infection, severity, and mortality [50,51]. A recent meta-analysis by Pal et al. including 10 observational studies and 3 randomized controlled trials (RCT), reported that VitD supplementation is associated with improved clinical outcomes (i.e., intensive care unit admission and/or mortality) of COVID-19 (OR 0.41; 95%CI 0.20–0.81), especially when VitD was administered after the diagnosis of COVID-19 (OR 0.35; 95%CI 0.14–0.85) [50]. The results from our current study also suggest that VitD supplementation might be beneficial in reducing hospitalisation of COVID-19 infection in the general population.

Strengths of our study include the relatively large number of participants and reliable outcome data prospectively collected through linkage with national registries. Also, the use of a central processing laboratory for biochemical assays, and a comprehensive questionnaire on measures of lifestyle completed at cohort entry, allowed us to adjust demographic and lifestyle risk factors and better analyse the link between our exposures and outcome. A major limitation of this study is that VitD levels and covariates were measured at baseline upon UK Biobank recruitment in 2006–2010, over a decade prior to the COVID-19 pandemic. As such, changes in 25(OH)D concentrations over time could not be captured, and misclassification of VitD status at the time of the pandemic is likely. Although 25(OH)D levels tend to remain relatively stable in adulthood, particularly after adjusting for seasonal variation, some degree of within-person variability over time is expected. This long-time gap introduces the potential for 'regression dilution' bias, which may lead to underestimation of true associations [36]. We were also unable to account for seasonal variation in VitD levels, as month data on the timing of blood collection were not available for participants included in our analysis. While a small subset of participants (n = 17,036) had a second measurement in 2012–2013, repeated measures were not available for most of the cohort. Moreover, although the UK Biobank is one of the few cohorts with a sufficiently large population to assess associations between VitD, ethnicity, and COVID-19 infection and hospitalisation, analyses within subgroups (e.g., cancer patients or specific ethnic groups) were limited by smaller sample sizes, which may have reduced statistical power and contributed to null findings in some cases. Another limitation of our study is the fact that study cohort is not completely representative of the UK population, in that participants of non-Caucasian ethnic backgrounds constitute only 6% of the UK biobank, tend to live in more socio-economically advantaged areas and are more likely to be older and female. Lastly, our measure of COVID-19 infection is based on testing data, and differential access to or uptake of testing may have introduced bias in measuring susceptibility to infection.

## Conclusion

This study investigated the association between VitD levels and COVID-19 outcomes, specifically infection and hospitalisation risks. Our results showed that VitD deficiency and insufficiency were strongly associated with an increased risk of hospitalisation, while no evidence was found of an increased risk of COVID-19 infection among those with insufficient or deficient VitD levels. Sensitivity analyses revealed that Black and Asian ethnic groups with lower VitD levels had a higher likelihood of COVID-19 infection, although this did not correspond to greater hospitalisation risks. These findings suggest that VitD and ethnicity may independently influence COVID-19 infection outcomes, rather than interacting to affect hospitalisation risks. Among participants with a prior cancer diagnosis, no significant associations were observed between VitD levels, ethnicity, and COVID-19 outcomes, potentially reflecting the unique immune challenges in this population. Overall, these results highlight the complex role of VitD in COVID-19 outcomes and the varying impact it has across different population subgroups.

## Supporting information

**S1 Table. Ethnicities included in UK Biobank.**
(DOCX)

**S2 Table. Stratified analyses for COVID-19 infection within the total population.** *Model A- Adjusted for Sex, Age at recruitment, Townsend Deprivation Index, overall health rating, BMI, and smoking status, with normal Vitamin D status as reference.
(DOCX)

**S3 Table. Stratified analyses for COVID-19 hospitalisation within the total population.** *Model A- Adjusted for Sex, Age at recruitment, Townsend Deprivation Index, overall health rating, BMI, and smoking status, with normal Vitamin D status as reference.
(DOCX)

**S4 Table. Stratified analyses for COVID-19 infection within the cancer population.** *Model A- Adjusted for Sex, Age at recruitment, Townsend Deprivation Index, overall health rating, BMI, and smoking status, with normal Vitamin D status as reference.
(DOCX)

**S5 Table. Stratified analyses for COVID-19 hospitalisation within the cancer population.** *Model A- Adjusted for Sex, Age at recruitment, Townsend Deprivation Index, overall health rating, BMI, and smoking status, with normal Vitamin D status as reference.
(DOCX)

## Author contributions

**Formal analysis:** Maria J. Monroy-Iglesias.

**Methodology:** Maria J. Monroy-Iglesias, Mieke Van Hemelrijck, Rathesh Thavarajah, Aida Santaolalla.

**Supervision:** Mieke Van Hemelrijck, Aida Santaolalla.

**Writing – original draft:** Maria J. Monroy-Iglesias, Rathesh Thavarajah.

**Writing – review & editing:** Mieke Van Hemelrijck, Debra H. Josephs, Kerri Beckmann, Sheeba Irshad, Sophia N. Karagiannis, Aida Santaolalla.

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
