## [Decision Letter · Decision Letter 0]

PONE-D-25-03098Effects of Vitamin D on COVID-19 Risk and Hospitalisation in the UK BiobankPLOS ONE

Dear Dr. Monroy Iglesias,

Thank you for submitting your manuscript to PLOS ONE. After careful consideration, we feel that it has merit but does not fully meet PLOS ONE’s publication criteria as it currently stands. Therefore, we invite you to submit a revised version of the manuscript that addresses the points raised during the review process.

We look forward to receiving your revised manuscript.

Kind regards,

Sandar Tin Tin

Academic Editor

PLOS ONE

Journal Requirements:

Additional Editor Comments:

In the analyses, please consider potential changes in Vitamin D levels over time (using a subsample of participants with repeat measurements available) as well as seasonal variations. 

Reviewers' comments:

Reviewer's Responses to Questions

**Comments to the Author**

1. Is the manuscript technically sound, and do the data support the conclusions?

Reviewer #1: Yes

Reviewer #2: No

2. Has the statistical analysis been performed appropriately and rigorously? 

Reviewer #1: Yes

Reviewer #2: I Don't Know

3. Have the authors made all data underlying the findings in their manuscript fully available?

Reviewer #1: Yes

Reviewer #2: Yes

4. Is the manuscript presented in an intelligible fashion and written in standard English?

Reviewer #1: Yes

Reviewer #2: Yes

5. Review Comments to the Author

Reviewer #1: This is a large study of vitamin D status and COVID-19 risk in the UK Biobank – it is not the first such study but it adds in data on a substantial subset with cancer and is well analysed and well written so adds helpfully to the literature. It has the well documented drawback that vitamin D levels are historical rather than current but this is well discussed – hopefully the authors might be able to provide more data on the utility of this analysis by correlating results over time in the patients who had more than one vitamin D level – see point 4 below.

A few other points need addressing.

1. The Abstract / Methods section should state when vitamin D levels were taken eg 2006-2013.

2. The Abstract / Results section is confusingly written – OR of less than 1.0 (0.97; 0.95) and OR of more than 1.0 (1.19 ; 1.36) are all presented as “positive associations” and OR of more than 1.0(1.50 etc) as “strong negative associations”.

I would suggest:

(a) Change second sentence eg to “Although unadjusted analyses showed a small increase in infection rates in association with vitamin D insufficiency (OR 1.07; 95%CI 1.04-1.10) and deficiency (OR 1.20; 95%CI 1.16-1.27), multivariate analysis conversely showed a borderline negative association between risk of COVID-19 infection in the whole cohort and vitD insufficiency (OR 0.97, 95%CI 0.94-1.00) or deficiency (OR 0.95; 95%CI 0.90-0.99).

(b) Change last sentence to: “Strong associations were found between vitamin D deficiency and increased risk of COVID-19 infection in Asian (OR 1.50; 95%CI 1.08-2.07), and Black (1.57; 95%CI 1.14-2.16) populations and also for increased risk of COVID-19 hospitalisation in the White (OR 1.44; 95% CI 1.25-1.66) population”.

(c) Move to the last sentence the reference to the cancer data eg “In the cancer population, no significant association was found between vitamin D status and either COVID-19 infection or hospitalisation”.

3. Introduction para 1 – it seems an odd omission not to mention obesity or ethnicity as risk factors for COVID-19 severity – moreover both of these factors are associated with vitamin D status.

4. Methods – vitamin D assessment – it is stated that a subset of the cohort (n=17,036) had a second vitamin D measurement some years after the first. It would be very helpful to know how well the two samplings correlated – perhaps people with previous low levels might have taken supplements – any data on this? And has any attempt been made to adjust vitamin D level for month/season of sampling?

5. Methods – COVID-19 assessment – should state when PCR status was checked – was there a fixed/same time point for all the cases and if so when? (eg “prior to …… date”.

6. Results line 2 “hospitalized with to COVID-19.”

7. Legends to Tables 3 to 6 repeat “smoking status”

8. Discussion – when citing other studies that have reported associations between COVID-19 risks and vitamin D status it would be helpful in each case to state what definition has been used for deficiency (eg 25(OH)D <25, <50, <75 nmol/l).

9. Discussion para 7 - the statement “The results from our current study also suggest that VitD

supplementation may be beneficial in reducing hospitalisation of COVID-19 infection in the general population.” Implies that you have data on supplementation – better would be “…. might be beneficial….”.

10. Discussion cites 3 previous studies of vitamin D/COVID-19 in the UK Biobank but a couple of others came up on a quick pubmed search: An observational and Mendelian randomisation study on vitamin D and COVID-19 risk in UK Biobank.

Li X, van Geffen J, van Weele M, Zhang X, He Y, Meng X, Timofeeva M, Campbell H, Dunlop M, Zgaga L, Theodoratou E.Sci Rep. 2021 Sep 14;11(1):18262. doi: 10.1038/s41598-021-97679-5.PMID: 34521884

Greater risk of severe COVID-19 in Black, Asian and Minority Ethnic populations is not explained by cardiometabolic, socioeconomic or behavioural factors, or by 25(OH)-vitamin D status: study of 1326 cases from the UK Biobank.

Raisi-Estabragh Z, McCracken C, Bethell MS, Cooper J, Cooper C, Caulfield MJ, Munroe PB, Harvey NC, Petersen SE.J Public Health (Oxf). 2020 Aug 18;42(3):451-460. doi: 10.1093/pubmed/fdaa095.PMID: 32556213

Reviewer #2: Despite the well-established links between VitD and COVID-19 (18-21)

19. Grant WB. An estimate of premature cancer mortality in the U.S. due to inadequate

doses of solar ultraviolet-B radiation. Cancer. 2002;94(6):1867-75.

20. Manson JE, Cook NR, Lee IM, Christen W, Bassuk SS, Mora S, et al. Vitamin D

Supplements and Prevention of Cancer and Cardiovascular Disease. N Engl J Med.

2019;380(1):33-44.

21. Vieth R. Critique of Public Health Guidance for Vitamin D and Sun Exposure in the

Context of Cancer and COVID-19. Anticancer Res. 2022;42(10):5027-34.

Comment: The statement in the text should separate the findings for COVID-19 from those for cancer.

Regarding Ref. 19, a better reference is

Muñoz A, Grant WB. Vitamin D and Cancer: An Historical Overview of the Epidemiology and Mechanisms. Nutrients. 2022 Mar 30;14(7):1448. doi: 10.3390/nu14071448.

A major limitation of this study is that exposures and covariates were measured at baseline upon recruitment into the UK Biobank in 2006 to 2010, thus potential changes on VitD cannot be accounted for. A small subset of the cohort had a second VitD measurement taken during the first repeat assessment visit between 2012 and 2013 (n=17,036).

Comment: The measurements are likely to be too old and seriously undercut the value of the analysis.

Also, were they seasonally adjusted?

Long time gaps are associated with “regression dilution”

Clarke R, Shipley M, Lewington S, Youngman L, Collins R, Marmot M, Peto R.

Underestimation of risk associations due to regression dilution in long-term follow-up of prospective studies.

Am J Epidemiol. 1999 Aug 15;150(4):341-53. doi: 10.1093/oxfordjournals.aje.a010013.

See, also, Figure 1 in

Muñoz A, Grant WB. Vitamin D and Cancer: An Historical Overview of the Epidemiology and Mechanisms. Nutrients. 2022 Mar 30;14(7):1448. doi: 10.3390/nu14071448.

Regarding season:

Hyppönen E, Power C.

Hypovitaminosis D in British adults at age 45 y: nationwide cohort study of dietary and lifestyle predictors.

Am J Clin Nutr. 2007 Mar;85(3):860-8. doi: 10.1093/ajcn/85.3.860.

Both Hastie et al and Lin et al found no associations between VitD levels and risk of COVID-19 hospitalisation and mortality (28, 29).

Comment: The patients in the Hastie study were in the hospital about 11 days after first symptoms of COVID-19. Since two of the important mechanisms of vitamin D are increasing cathelicidin (LL-37) concentrations, which can reduce the viability of viruses and bacteria, and reduced production of pro-inflammatory cytokines, which can lead to the cytokine storm, are most beneficial early in the disease progression. Vitamin D can also reduce risk of death from ensuing pneumonia.

It should also be noted that acute inflammatory diseases reduce 25OHD concentrations. See

Smolders, J.; van den Ouweland, J.; Geven, C.; Pickkers, P.; Kox, M. Letter to the Editor: Vitamin D deficiency in COVID-19: Mixing up cause and consequence. Metabolism 2021, 115, 154434

And search Google Scholar for papers that cited this letter.

Significant digits. The general rule is that no more non-zero digits should be given than are justified by the uncertainty of the value.

See "Too many digits: the presentation of numerical data"

https://www.ncbi.nlm.nih.gov/pmc/articles/PMC4483789/

If the uncertainty (or difference when comparing numbers) is greater than about 7%, only two non-zero digits are justified.

P values should be given to two decimal places unless the first two are 00 or the number lies between 0.045 and 0.054. If the first two are 00, then only one non-zero digit can be given.

Thus, p values in the supplementary file should be adjusted

In Table 2

Mean (SD) 27.84 (4.82)

Should be

Mean (SD) 28 (5)

Please review all numbers in abstract, text, tables, and figures and adjust accordingly.

6. PLOS authors have the option to publish the peer review history of their article (what does this mean? ). If published, this will include your full peer review and any attached files.

**Do you want your identity to be public for this peer review?** For information about this choice, including consent withdrawal, please see our Privacy Policy .

Reviewer #1: No

Reviewer #2: **Yes: ** William B. Grant

---

## [Author Response · Author response to Decision Letter 1]

22 May 2025

Response to reviewers

AUTHORS’ REPLY: We would like to sincerely thank the reviewers for their thoughtful and constructive feedback. Their insights have been invaluable in strengthening our manuscript, and we appreciate the time they took to carefully review our work. Our response to reviewers is outlined below:

Reviewer #1 (Remarks to the author):

This is a large study of vitamin D status and COVID-19 risk in the UK Biobank – it is not the first such study but it adds in data on a substantial subset with cancer and is well analysed and well written so adds helpfully to the literature. It has the well documented drawback that vitamin D levels are historical rather than current but this is well discussed – hopefully the authors might be able to provide more data on the utility of this analysis by correlating results over time in the patients who had more than one vitamin D level – see point 4 below.

A few other points need addressing.

1. The Abstract / Methods section should state when vitamin D levels were taken eg 2006-2013.

AUTHORS’ REPLY: Thank you for this suggestion. We have clarified in the Abstract Methods that all individuals included in the study had at least one serum vitamin D measurement taken at baseline (2006–2010). As this was the relevant measurement for cohort inclusion, we have specified this timeframe in the abstract.

2. The Abstract / Results section is confusingly written – OR of less than 1.0 (0.97; 0.95) and OR of more than 1.0 (1.19 ; 1.36) are all presented as “positive associations” and OR of more than 1.0(1.50 etc) as “strong negative associations”.

I would suggest:

(a) Change second sentence eg to “Although unadjusted analyses showed a small increase in infection rates in association with vitamin D insufficiency (OR 1.07; 95%CI 1.04-1.10) and deficiency (OR 1.20; 95%CI 1.16-1.27), multivariate analysis conversely showed a borderline negative association between risk of COVID-19 infection in the whole cohort and vitD insufficiency (OR 0.97, 95%CI 0.94-1.00) or deficiency (OR 0.95; 95%CI 0.90-0.99).

(b) Change last sentence to: “Strong associations were found between vitamin D deficiency and increased risk of COVID-19 infection in Asian (OR 1.50; 95%CI 1.08-2.07), and Black (1.57; 95%CI 1.14-2.16) populations and also for increased risk of COVID-19 hospitalisation in the White (OR 1.44; 95% CI 1.25-1.66) population”.

(c) Move to the last sentence the reference to the cancer data eg “In the cancer population, no significant association was found between vitamin D status and either COVID-19 infection or hospitalisation”.

AUTHORS’ REPLY: Thank you for pointing this out and for your suggestions. We agree that the language and order in the original abstract may have caused confusion. We have revised the Results section of the abstract to more clearly describe the direction of associations. The revised results section of our abstract now reads like the below.

“Results

Of 151,543 participants, 21,396 tested positive for COVID-19. Of 24,400 individuals with cancer, 2,608 tested positive. In the total cohort, VitD insufficiency (Adjusted Odds Ratio (aOR) 0.97, 95% Confidence Interval (CI) 0.94-1.00) and deficiency (aOR 0.95, 95%CI 0.90-0.99) were associated with slightly lower odds of COVID-19 infection. In contrast, both VitD insufficiency (aOR 1.19, 95%CI 1.08-1.31) and deficiency (aOR 1.36, 95%CI 1.19-1.56) were associated with higher odds of COVID-19 hospitalisation. Among Asian (aOR 1.50; 95%CI 1.08-2.07) and Black (aOR 1.57; 95%CI 1.14-2.16) participants, VitD deficiency was associated with higher odds of COVID-19 infection. Among White participants, VitD insufficiency was associated with slightly lower odds of COVID-19 infection (aOR 0.97; 95%CI 0.86-0.95), while both VitD insufficiency (aOR 1.19; 95%CI 1.08-1.32) and deficiency (aOR 1.44; 95%CI 1.25-1.66) were associated with increased odds of hospitalisation. In the cancer population, vitamin D deficiency was associated with higher odds of infection only among Black participants (aOR 3.50; 95%CI 1.22-10.01); no other associations were observed.”

3. Introduction para 1 – it seems an odd omission not to mention obesity or ethnicity as risk factors for COVID-19 severity – moreover both of these factors are associated with vitamin D status.

AUTHORS’ REPLY: We thank the reviewer for this helpful suggestion. We agree that it is important to mention obesity and ethnicity as risk factors for COVID-19 severity, particularly given their association with vitamin D status. We have updated the first paragraph of the Introduction to include obesity and belonging to certain ethnic groups as established risk factors for COVID-19 infection and severity. In addition, we have edited the second paragraph to improve the flow and explicitly link these factors to both vitamin D deficiency and COVID-19 outcomes. Additional references have been incorporated to support these revisions.

4. Methods – vitamin D assessment – it is stated that a subset of the cohort (n=17,036) had a second vitamin D measurement some years after the first. It would be very helpful to know how well the two samplings correlated – perhaps people with previous low levels might have taken supplements – any data on this? And has any attempt been made to adjust vitamin D level for month/season of sampling?

AUTHORS’ REPLY: Thank you very much for this thoughtful and important comment. A subset of participants (n=17,036) had two vitamin D measurements taken during the initial baseline visit and again at the first repeat assessment between 2012 and 2013. In our initial analyses, we assessed the correlation between these two measurements and found them to be highly correlated with minimal variability over time. Based on this, we chose to use the median of the two measurements to provide a more stable and representative estimate of typical vitamin D status for each individual, helping to reduce the influence of any short-term fluctuations or outliers. Unfortunately, we no longer have access to the original dataset to provide specific correlation coefficients, but this approach was justified by the observed consistency at the time of analysis. We have added the following clarifying sentence to the methods section to explain this:

"A small subset of the cohort had a second VitD measurement taken during the first repeat assessment visit between 2012 and 2013 (n=17,036). For individuals with two measurements, the average VitD level was calculated. As the two measurements were highly correlated, averaging them provided a stable estimate of typical VitD status."

Regarding adjustment for seasonality, we appreciate this important point and did consider adjusting for the season when vitamin D was measured. Unfortunately, this was not possible because data on the specific month or season of blood collection were not available for the participants included in our analysis. Unlike other biomarkers collected across the entire UK Biobank cohort, vitamin D was measured only in a subset, and relevant timing data were not accessible for this group. We acknowledge that season is an important factor influencing vitamin D levels and have added the following statement in the manuscript to acknowledge this limitation:

"We were also unable to account for seasonal variation in vitamin D levels, as data on the timing of blood collection were not available for participants included in our analysis."

We hope this clarifies these points and appreciate your helpful suggestions.

5. Methods – COVID-19 assessment – should state when PCR status was checked – was there a fixed/same time point for all the cases and if so when? (eg “prior to …… date”.

AUTHORS’ REPLY: Thank you for your helpful comment. We have clarified in the Methods section (COVID-19 assessment) that PCR test results included in our study were collected between March and October 2020. All analyses used the earliest positive PCR test result within this period. This has been added for clarity.

6. Results line 2 “hospitalized with to COVID-19.”

7. Legends to Tables 3 to 6 repeat “smoking status”

AUTHORS’ REPLY: Thank you for noting those two errors in our manuscript, we have now corrected them.

8. Discussion – when citing other studies that have reported associations between COVID-19 risks and vitamin D status it would be helpful in each case to state what definition has been used for deficiency (eg 25(OH)D <25, <50, <75 nmol/l).

AUTHORS’ REPLY: Thank you for this helpful suggestion. We have now added the vitamin D cut-off values used to define deficiency and insufficiency for all cited studies where available.

9. Discussion para 7 - the statement “The results from our current study also suggest that VitD

supplementation may be beneficial in reducing hospitalisation of COVID-19 infection in the general population.” Implies that you have data on supplementation – better would be “…. might be beneficial….”.

AUTHORS’ REPLY: Thank you for this suggestion, it really improves the accuracy of the discussion. We have amended based on your suggestion.

10. Discussion cites 3 previous studies of vitamin D/COVID-19 in the UK Biobank but a couple of others came up on a quick pubmed search:

An observational and Mendelian randomisation study on vitamin D and COVID-19 risk in UK Biobank.

Li X, van Geffen J, van Weele M, Zhang X, He Y, Meng X, Timofeeva M, Campbell H, Dunlop M, Zgaga L, Theodoratou E.Sci Rep. 2021 Sep 14;11(1):18262. doi: 10.1038/s41598-021-97679-5.PMID: 34521884

Greater risk of severe COVID-19 in Black, Asian and Minority Ethnic populations is not explained by cardiometabolic, socioeconomic or behavioural factors, or by 25(OH)-vitamin D status: study of 1326 cases from the UK Biobank.

Raisi-Estabragh Z, McCracken C, Bethell MS, Cooper J, Cooper C, Caulfield MJ, Munroe PB, Harvey NC, Petersen SE.J Public Health (Oxf). 2020 Aug 18;42(3):451-460. doi: 10.1093/pubmed/fdaa095.PMID: 32556213

AUTHORS’ REPLY: Thank you for bringing these additional studies to our attention. We have now incorporated the findings from both Li et al. (2021) and Raisi-Estabragh et al. (2020) into the Discussion section where relevant, to provide a more comprehensive overview of the existing literature using UK Biobank data on vitamin D and COVID-19 outcomes.

Reviewer #2 (Remarks to the author):

1. Despite the well-established links between VitD and COVID-19 (18-21)

19. Grant WB. An estimate of premature cancer mortality in the U.S. due to inadequate

doses of solar ultraviolet-B radiation. Cancer. 2002;94(6):1867-75.

20. Manson JE, Cook NR, Lee IM, Christen W, Bassuk SS, Mora S, et al. Vitamin D

Supplements and Prevention of Cancer and Cardiovascular Disease. N Engl J Med.

2019;380(1):33-44.

21. Vieth R. Critique of Public Health Guidance for Vitamin D and Sun Exposure in the

Context of Cancer and COVID-19. Anticancer Res. 2022;42(10):5027-34.

Comment: The statement in the text should separate the findings for COVID-19 from those for cancer.

Regarding Ref. 19, a better reference is

Muñoz A, Grant WB. Vitamin D and Cancer: An Historical Overview of the Epidemiology and Mechanisms. Nutrients. 2022 Mar 30;14(7):1448. doi: 10.3390/nu14071448.

AUTHORS’ REPLY: Thank you for pointing out the discrepancies in the references. We have replaced the (19) with the recommended reference (Muñoz & Grant, Nutrients, 2022) and revised the paragraph to ensure clarity and accuracy. This also prompted us to refine the wording of the entire paragraph to improve its flow and better reflect the evidence base (see revised text below):

‘Although associations between VitD status and COVID-19 outcomes have been examined in the general population, there remains a notable gap in research focusing on individuals with cancer. Given VitD’s potential protective role in cancer—through mechanisms such as immune modulation and regulation of cell growth (18-21)—it is important to consider how VitD status may influence COVID-19 outcomes in this group. Understanding the potential link between VitD and COVID-19 within the cancer population could inform clinical practice and public health strategies aimed at mitigating COVID-19 morbidity and mortality in this vulnerable group.’

2. A major limitation of this study is that exposures and covariates were measured at baseline upon recruitment into the UK Biobank in 2006 to 2010, thus potential changes on VitD cannot be accounted for. A small subset of the cohort had a second VitD measurement taken during the first repeat assessment visit between 2012 and 2013 (n=17,036).

Comment: The measurements are likely to be too old and seriously undercut the value of the analysis.

Also, were they seasonally adjusted?

Long time gaps are associated with “regression dilution”

Clarke R, Shipley M, Lewington S, Youngman L, Collins R, Marmot M, Peto R.

Underestimation of risk associations due to regression dilution in long-term follow-up of prospective studies.

Am J Epidemiol. 1999 Aug 15;150(4):341-53. doi: 10.1093/oxfordjournals.aje.a010013.

See, also, Figure 1 in

Muñoz A, Grant WB. Vitamin D and Cancer: An Historical Overview of the Epidemiology and Mechanisms. Nutrients. 2022 Mar 30;14(7):1448. doi: 10.3390/nu14071448.

Regarding season:

Hyppönen E, Power C.

Hypovitaminosis D in British adults at age 45 y: nationwide cohort study of dietary and lifestyle predictors.

Am J Clin Nutr. 2007 Mar;85(3):860-8. doi: 10.1093/ajcn/85.3.860.

AUTHORS’ REPLY: Thank you for your helpful comment highlighting the potential impact of outdated vitamin D measurements and the issue of regression dilution. We agree that the long time gap between baseline vitamin D assessment and the COVID-19 pandemic is a key limitation. In response, we have revised the limitations section of the manuscript to more clearly reflect this concern and have incorporated language to highlight the importance of this limitation.

Regarding adjustment for seasonality, we appreciate the references provided and acknowledge that seasonal variation is an important determinant of vitamin D status. Unfortunately, data on the specific month or season of blood collection were not available for participants included in our analysis, which prevented us from adjusting for this factor. We have added a statement in the manuscript to explicitly acknowledge this limitation.

‘Strengths of our study include the relatively large number of participants and reliable outcome data prospectively collected through linkage with national registries. Also, the use of a central processing laboratory for biochemical assays, and a comprehensive questionnaire on measures of lifestyle completed at cohort entry, allowed us to adjust demographic and lifestyle risk factors and better analyse the link between our exposures and outcome. A major limitation of this study is that VitD levels and covariates were measured at baseline upon UK Biobank recruitment in 2006 to 2010, over a decade prior to the COVID-19 pandemicAs such, changes in 25(OH)D concentrations over time could not be captured, and misclassification of VitD status at the time of the pandemic is likely. Although 25(OH)D levels tend to remain relatively stable in adulthood, particularly after adjusting for seasonal variation, some degree of within-person variability over time is expected. This long-time gap introduces the potential for ‘regression dilution’ bias, which may lead to underestimation of true associations (36). We were also unable to account for seasonal variation in vitamin D levels, as month data on the timing of blood collection were not available for participants included in our analysis. While a small subset of participants (n=17,036) had a second measurement in 2012–2013, repeated measures were not available for most of the cohort. Moreover, although the UK Biobank is one of the few cohorts with a sufficiently large population to assess associations between VitD, ethnicity, and COVID-19 infection and hospitalisation, analyses within subgroups (e.g., cancer patients or specific ethnic groups) were limited by smaller sample sizes, which may have reduced statistical power and contributed to null findings in some cases. Another limitation of our study is the fact that study cohort is not completely representative of the UK population, in that participants

---

## [Decision Letter · Decision Letter 1]

PONE-D-25-03098R1Effects of Vitamin D on COVID-19 Risk and Hospitalisation in the UK BiobankPLOS ONE

Dear Dr. Monroy Iglesias,

Thank you for submitting your manuscript to PLOS ONE. After careful consideration, we feel that it has merit but does not fully meet PLOS ONE’s publication criteria as it currently stands. Therefore, we invite you to submit a revised version of the manuscript that addresses the points raised during the review process.

We look forward to receiving your revised manuscript.

Kind regards,

Sandar Tin Tin

Academic Editor

PLOS ONE

Journal Requirements:

Reviewers' comments:

Reviewer's Responses to Questions

**Comments to the Author**

1. If the authors have adequately addressed your comments raised in a previous round of review and you feel that this manuscript is now acceptable for publication, you may indicate that here to bypass the “Comments to the Author” section, enter your conflict of interest statement in the “Confidential to Editor” section, and submit your "Accept" recommendation.

Reviewer #1: All comments have been addressed

Reviewer #2: (No Response)

2. Is the manuscript technically sound, and do the data support the conclusions?

Reviewer #1: Yes

Reviewer #2: Yes

3. Has the statistical analysis been performed appropriately and rigorously? 

Reviewer #1: Yes

Reviewer #2: Yes

4. Have the authors made all data underlying the findings in their manuscript fully available?

Reviewer #1: Yes

Reviewer #2: Yes

5. Is the manuscript presented in an intelligible fashion and written in standard English?

Reviewer #1: Yes

Reviewer #2: Yes

6. Review Comments to the Author

Reviewer #1: All comments satisfactorily addressed

Reviewer #2: .”However, we were unable to find the reference in Hastie et al. regarding patients being hospitalized about 10 days after symptom onset. If you happen to recall where that came from, we’d be happy to take another look, but for now we’ve left that detail out.

Re: Hastie. My mistake.

However, this article may be of interest.

Baseline clinical features of COVID-19 patients, delay of hospital admission and clinical outcome: A complex relationship.

Dananché C, Elias C, Hénaff L, Amour S, Kuczewski E, Gustin MP, Escuret V, Saadatian-Elahi M, Vanhems P. PLoS One. 2022 Jan 7;17(1):e0261428. doi: 10.1371/journal.pone.0261428.

Of possible interest

Vitamin D Deficiency Is Associated With Higher Hospitalization Risk From COVID-19: A Retrospective Case-control Study.

Jude EB, Ling SF, Allcock R, Yeap BXY, Pappachan JM. J Clin Endocrinol Metab. 2021 Oct 21;106(11):e4708-e4715. doi: 10.1210/clinem/dgab439.

7. Jolliffe DA, Griffiths CJ, Martineau AR. Vitamin D in the prevention of acute

respiratory infection: systematic review of clinical studies. J Steroid Biochem Mol Biol.

2013;136:321-9.

Comment: This reference is outdated. See

Vitamin D supplementation to prevent acute respiratory infections: systematic review and meta-analysis of stratified aggregate data.

Jolliffe DA, Camargo CA Jr, Sluyter JD, Aglipay M, Aloia JF, Bergman P, Bischoff-Ferrari HA, Borzutzky A, Bubes VY, Damsgaard CT, Ducharme FM, Dubnov-Raz G, Esposito S, Ganmaa D, Gilham C, Ginde AA, Golan-Tripto I, Goodall EC, Grant CC, Griffiths CJ, Hibbs AM, Janssens W, Khadilkar AV, Laaksi I, Lee MT, Loeb M, Maguire JL, Majak P, Manaseki-Holland S, Manson JE, Mauger DT, Murdoch DR, Nakashima A, Neale RE, Pham H, Rake C, Rees JR, Rosendahl J, Scragg R, Shah D, Shimizu Y, Simpson-Yap S, Kumar GT, Urashima M, Martineau AR. Lancet Diabetes Endocrinol. 2025 Apr;13(4):307-320. doi: 10.1016/S2213-8587(24)00348-6

7. PLOS authors have the option to publish the peer review history of their article (what does this mean? ). If published, this will include your full peer review and any attached files.

**Do you want your identity to be public for this peer review?** For information about this choice, including consent withdrawal, please see our Privacy Policy .

Reviewer #1: No

Reviewer #2: **Yes: ** William B. Grant

---

## [Author Response · Author response to Decision Letter 2]

25 Jun 2025

AUTHORS’ REPLY: Thank you very much for your incredibly useful references and insightful comments. We have updated the outdated references with those you kindly provided and included the study on delays between symptom onset and hospital admission into our introduction (Vitamin D supplementation to prevent acute respiratory infections: systematic review and meta-analysis of stratified aggregate data. Jolliffe DA) and discussion (Baseline clinical features of COVID-19 patients, delay of hospital admission and clinical outcome: A complex relationship. Dananché C & Vitamin D Deficiency Is Associated With Higher Hospitalization Risk From COVID-19: A Retrospective Case-control Study. Jude EB).

In particular, we have revised the following paragraph of the discussion to incorporate these points more clearly:

“Although epidemiological evidence on the link between VitD and COVID-19 severity remains inconsistent, there are strong biological reasons to support a potential connection. VitD plays a crucial regulatory role in both acquired and innate immunity, and higher levels have been associated with a reduced risk of the cytokine storm and subsequent acute respiratory distress syndrome (ARDS), a leading cause of COVID-19 mortality (34, 35). Two key mechanisms by which VitD may exert protective effects include the upregulation of antimicrobial peptides such as cathelicidin (LL-37), which reduce viral and bacterial viability, and the suppression of pro-inflammatory cytokines that contribute to disease progression. These effects are likely most beneficial early in the course of infection (36). Importantly, clinical studies have highlighted the complexity of COVID-19 progression, including evidence that shows delays between symptom onset and hospital admission which may in turn influence patient outcomes (37, 38). Furthermore, acute inflammation itself may reduce circulating 25(OH)D concentrations, complicating the interpretation of VitD measurements taken during or shortly after illness onset (39, 40). In our study, no significant association was observed between VitD levels and COVID-19 infection or hospitalization among participants with a prior cancer diagnosis. This lack of association may be due to immune suppression from cancer treatments, such as chemotherapy, which could blunt the immune-modulating effects of VitD (41). Alternatively, the chronic inflammation and immune dysfunction commonly seen in cancer patients might overshadow any potential protective benefits of adequate VitD levels (42). Further studies are needed to disentangle the interplay between cancer, immune function, and VitD in the context of COVID-19.”

---

## [Editor Report · Decision Letter 2]

Effects of Vitamin D on COVID-19 Risk and Hospitalisation in the UK Biobank

PONE-D-25-03098R2

Dear Dr. Monroy Iglesias,

We’re pleased to inform you that your manuscript has been judged scientifically suitable for publication and will be formally accepted for publication once it meets all outstanding technical requirements.

Kind regards,

Sandar Tin Tin

Academic Editor

PLOS ONE

---

## [Editor Report · Acceptance letter]

PONE-D-25-03098R2

PLOS ONE

Dear Dr. Monroy Iglesias,

I'm pleased to inform you that your manuscript has been deemed suitable for publication in PLOS ONE. Congratulations! Your manuscript is now being handed over to our production team.

Kind regards,

on behalf of

Dr. Sandar Tin Tin

Academic Editor

PLOS ONE